# West Nile Virus, an Underdiagnosed Cause of Acute Fever of Unknown Origin and Neurological Disease among Hospitalized Patients in South Africa

**DOI:** 10.3390/v15112207

**Published:** 2023-11-02

**Authors:** Caitlin MacIntyre, Carla Lourens, Adriano Mendes, Maryke de Villiers, Theunis Avenant, Nicolette M. du Plessis, Fabian H. Leendertz, Marietjie Venter

**Affiliations:** 1Zoonotic Arbo- and Respiratory Virus Program, Department of Medical Virology, Faculty of Health Sciences, University of Pretoria, Pretoria 0031, South Africa; cdm.macintyre@gmail.com (C.M.); clourens101@gmail.com (C.L.); adrianomendes288@gmail.com (A.M.); 2Department of Internal Medicine, Kalafong Provincial Tertiary Hospital, Faculty of Health Sciences, University of Pretoria, Pretoria 0031, South Africa; maryke.devilliers@up.ac.za; 3Department of Pediatrics, Kalafong Provincial Tertiary Hospital, Faculty of Health Sciences, University of Pretoria, Pretoria 0031, South Africa; theunis.avenant@up.ac.za (T.A.); nicolette.duplessis@up.ac.za (N.M.d.P.); 4Helmholtz Institute for One Health and University of Greifswald, 17489 Greifswald, Germany; fabian.leendertz@helmholtz-hioh.de

**Keywords:** West Nile virus, flavivirus, arbovirus, mosquito-borne, hospitalized patients, acute fever, neurological disease, IgM, serology

## Abstract

West Nile virus (WNV), a mosquito-borne flavivirus, is endemic to South Africa. However, its contribution to acute febrile and neurological disease in hospitalized patients in South Africa is unknown. This study examined two patient cohorts for WNV using molecular testing and IgM serology with confirmation of serological results by viral neutralization tests (VNT) to address this knowledge gap. Univariate analysis was performed using collected demographic and clinical information to identify risk factors. In the first cohort, 219 cerebrospinal fluid (CSF) specimens from patients with acute neurological disease in Gauteng hospitals collected in January to June 2017 were tested for WNV. The study identified WNV in 8/219 (3.65%, 95.00% CI (1.59–7.07)) patients with unsolved neurological infections. The second cohort, from 2019 to 2021, included 441 patients enrolled between January and June with acute febrile or neurological disease from urban and rural sites in Gauteng and Mpumalanga provinces. West Nile virus was diagnosed in 40/441 (9.07%, 95.00% CI (6.73–12.12)) of patients, of which 29/40 (72.50%, 95.00% CI (56.11–85.40)) had neurological signs, including headaches, encephalitis, meningitis, and acute flaccid paralysis (AFP). Notably, most of the cases were identified in children although adolescents and senior adults had a significantly higher risk of testing WNV positive. This suggests a previously underestimated disease burden and that WNV might be underrecognized as a cause of febrile and neurological diseases in hospitalized patients in South Africa, especially in children. This emphasizes the importance of further research and awareness regarding arboviruses of public health concern.

## 1. Introduction

West Nile virus (WNV) is a vector-borne arbovirus (Family: *Flaviviridae*, Genus: *Orthoflavivirus*) first identified in a febrile woman from Uganda in 1937 [1]. Typically, only 20.00% of infections manifest clinically, usually as a mild febrile illness [2]. Neurological complications can occur in 10.00% of clinically ill patients [2]. Of these, the case fatality rate (CFR) is around 10.00% but is significantly higher in the elderly, immunocompromised or those with WNV encephalitis or poliomyelitis [3]. Diagnosing WNV infections is challenging due to the non-specific flu-like symptoms, and the neurological presentations are indistinguishable from other pathogens [3]. Due to the fleeting viremia observed in WNV infection, the detection of WNV-specific IgM antibodies in serum or CSF is the gold standard of diagnosis of acute infections [2]. As antibodies do not normally cross the blood–brain barrier, the detection of WNV IgM in the CSF is a marker of neuroinvasive disease, as IgM antibodies can only permeate the blood–brain barrier due to inflammation [4]. However, the persistence of WNV IgM antibodies complicates the diagnosis of acute infections. Studies have shown that WNV IgM antibodies can persist in serum from anywhere between two months up to 8 years after infection [5]. Thus, they may not be a true marker of acute infection [5]. However, WNV IgM antibodies seem to persist for less time in CSF (up to 7 months), making CSF preferential for diagnosing acute neurological infections over serum [4]. To complicate diagnosis further, a positive serological test might be due to cross-reactive antibodies from an infection with a related flavivirus [6]. Therefore, accurate diagnosis of acute WNV infection should consider both clinical symptoms and molecular and/or serological results that have been confirmed with WNV VNTs. 

Outbreaks of WNV in the human population before the early 1990s were sporadic, thought to be self-limiting, and neurological infections were uncommon [7]. Information on the epidemiology and ecology of the disease was based primarily on work from the Mediterranean and Egypt in the 1950s [7]. The first well-established link between WNV and neurological disease in humans was identified in Israel in 1957 following an outbreak in a nursing home, whereby encephalitis developed in one-third of the infected patients [8]. One of the largest WNV outbreaks ever recorded occurred in 1974 in the Karoo and Northern Cape provinces of South Africa (SA), where over 10,000 febrile cases were reported, however, no neurological cases were noted [9]. The subsequent expansion of WNV in West and Central Europe, and its emergence and continuous spread across the Americas since 2000 has made WNV the most widespread cause of arboviral encephalitis worldwide [10].

Currently, nine WNV lineages are proposed but lineages 1 and 2 have been associated with outbreaks [11]. Lineage clade 1a circulates in North Africa, Asia, Europe, and the Americas, and clade 1b is confined to Australia [10]. Lineage 2 was thought to be geographically confined to sub-Saharan Africa and Madagascar but emerged in Europe in 2004, following the identification of an isolate in Hungary obtained from an encephalitic goshawk [10]. Lineage 2 has since swiftly expanded in Central Europe and the Mediterranean region, with outbreaks characterized by unprecedented numbers of neurological disease in humans and animals [10]. Most recently, lineage 2 was identified in mosquitoes and humans for the first time in the Netherlands [12]. In Europe, during the 2022 WNV transmission season, over 965 human cases of WNV, and 73 deaths, have been reported [13]. These recent outbreaks have shifted the focus from lineage 1 to lineage 2 as being of high public health concern [14].

Despite being endemic in SA, only 5–15 human WNV cases are reported annually, although this is probably an underestimation due to a lack of clinical awareness [9]. A single study focusing on patients experiencing acute febrile or neurological disease in the Tshwane region of SA from 2008 to 2009 identified serum-neutralizing antibodies (nAbs) to WNV in 40/206 (19.42%) patients, while 4/15 (26.67%) CSF specimens had nAbs [15]. However, IgM antibodies could only be confirmed in 2/206 (0.97%) serum specimens due to the limited specimen volume available and 1/190 (0.53%) were PCR positive. In SA, no study has investigated risk factors for WNV infection, nor has the burden of the disease been investigated in hospitalized patients experiencing acute fever of unknown cause with or without neurological symptoms. To address this gap, we sought to investigate the association of WNV with neurological disease using molecular and IgM serodiagnosis coupled with WNV VNT for confirmation in two patient cohorts. The first cohort was derived from CSF specimens collected from patients presenting to Gauteng hospitals with neurological disease of unknown etiology in 2017. The second cohort focused on patients presenting with acute febrile or neurological disease at three hospitals in the Gauteng and Mpumalanga provinces from 2019 to 2021. These provinces have temperate and sub-tropical climates, respectively. Demographic and socioeconomic risk factors as well as clinical presentation for WNV infection were investigated. This allowed us to investigate the contribution of WNV in unsolved cases of fever, with or without neurological symptoms, in hospitalized patients in the Gauteng and Mpumalanga provinces of SA. 

## 2. Materials and Methods

### 2.1. Ethical Statement

The University of Pretoria’s Faculty of Health Sciences ethics committee (100/2017 and 101/2017) approved the study. Informed consent was obtained as outlined by the African Network for Improved Diagnostics, Epidemiology, and Management of Common Infections Agents (ANDEMIA) study [16]. 

### 2.2. Study Design and Sentinel Sites

Two patient cohorts were used in this study. The first used archived CSF specimens sampled from patients suffering acute neurological disease at 10 public sector hospitals in Gauteng Province (see Appendix A for a map of the hospitals) that were submitted to the National Health Laboratory Service (NHLS) for viral diagnoses (hereinafter described as the neurological cohort). Cases were de-identified, assigned study numbers, and entered into a database containing no identifying information. Basic demographic data and diagnostic findings were recorded through the NHLS TrakCare database. Only specimens collected between January and June 2017 were tested for WNV, as this covers the South African mid-summer to autumn months, when the availability of mosquito breeding sites and vectorial capacity increases due to the warmer and wetter weather conditions [17]. This seasonality has also been confirmed through sentinel WNV surveillance in horses over 8 years [18]. 

The second cohort performed prospective sampling as part of the larger ANDEMIA study and consisted of two sentinel sites in SA, namely Kalafong Provincial Tertiary Hospital, which is an urban teaching hospital in the Gauteng province, and the rural Matikwane and Mapulaneng hospitals, in the Mpumalanga province (see Appendix A for a map of the hospitals) [16]. Prospective febrile and neurological disease surveillance was conducted from January 2019 to December 2021 at both sentinel sites in patients meeting the AFDUC case definition with or without acute neurological symptoms (hereinafter described as the AFDUC cohort). The case definition aimed at identifying AFDUC patients with arboviral infections [19]. Patients were subsequently enrolled under informed consent by surveillance officers, and a case investigation form was completed to capture basic demographic, socioeconomic, and clinical data. A one-month follow-up was performed where possible to determine the recovery rate. Cases were de-identified, assigned study numbers, and entered into a database containing no identifying information. 

### 2.3. Sample Processing and RNA Extraction

For the neurological cohort, only CSF specimens were obtained and processed. For each patient enrolled into the AFDUC cohort, EDTA (ethylene diamine-tetra acetic acid) blood was taken. In some cases, serum blood was collected as well. CSF was only taken if deemed necessary by medical staff as part of the standard of care. CSF and EDTA blood were used for nucleic acid extractions using the Qiagen RNA Viral mini kit (Qiagen, Valencia, USA). For serological testing, serum was preferentially used or EDTA was centrifuged at 2000× *g* for 15 min to separate the plasma. If a single patient had both blood and CSF specimens collected, CSF was preferentially tested for the presence of IgM antibodies due to it being a marker of acute neuroinvasive disease. Therefore, a single specimen was tested for each patient. All methods using commercial kits were carried out according to manufactures recommendations.

### 2.4. Multi-Pathogen Detection in the AFDUC Cohort

Routine molecular surveillance for ANDEMIA specimens was performed using the Chipron LCD macroarray assay (Chipron GmbH, Berlin, Germany), which is a multiplex PCR designed to simultaneously detect 30 pathogens, including WNV and other flaviviruses [20].

### 2.5. Flavivirus Detection in the Neurological Cohort 

Flavivirus screening using real-time RT-PCR, subsequent Sanger sequencing and phylogenetic analysis was performed as described previously [15] on the 2017 CSF specimens in the neurological cohort. For phylogenetic comparison to other South African strains, WNV sequences obtained from mosquitoes [17] and animals suffering neurological disease in SA [21] were included. 

### 2.6. Euroimmun WNV IgM ELISA and Neutralization Tests

Serum/plasma and CSF specimens from both cohorts were screened for the presence of anti-WNV IgM antibodies using the Euroimmun WNV IgM ELISA kit (Euroimmun, Lübeck, Germany). Sera or plasma was diluted at 1:101 with the sample buffer as recommended, while CSF was diluted at 1:2. Due to the cross-reactive nature of flaviviruses, all IgM positive or inconclusive results were confirmed using a WNV VNT with 10^5^ TCID50 U/mL (50.00% tissue culture infectious dose) of WNV culture (HS101/08, accession number JN393308. passage 6) as previously described [22]. Briefly, VNTs were performed in flat-bottomed 96 well plates. Two-fold dilutions (1: 8, 1: 16, 1:3 2) of test sera/plasma/CSF were prepared using Earle’s minimum essential medium (EMEM) containing 2.00% fetal calf serum and 100 μL was added in duplicate to the plate. To this, 100 μL of 100 TCID50/mL WNV prepared in 2.00% EMEM was added. To prepare the back titration, 10-fold serial dilutions of the 100TCID50 stock were created, until 0 TCID50 was reached, and 100 μL of these dilutions were added in quadruplicate to the plate. The plates were placed in humidity chambers and incubated for 1 h at 37 °C with 5.00% CO_2_. After which, 80 μL of 8 × 10^5^ Vero cells/mL were added to each well. Negative control wells consisted of Vero cells and EMEM only. The plates were incubated for 4–7 days at 37 °C with 5.00% CO_2_. Only once the back titration reads 100.00% cytopathic effect (CPE) at 100 TCID50, 75.00% CPE at 10 TCID50 and 25.00% CPE at 1 TCID50 and 0 CPE at 0 TCID50 was the plate was read. Specimens showing VNT titers ≥ 1: 16 were considered positive for WNV. To rule out further cross-reactions, any WNV IgM positive cases that could not be confirmed by WNV VNTs, were tested for neutralizing antibodies against the following co-circulating flaviviruses: Wesselsbron virus, Banzi virus, Bagaza virus, and Usutu virus. See Appendix A for details of the virus isolates used. 

### 2.7. Data Analysis

The patients’ ages were categorized into four groups based on the Neural Network using the Face and Gesture Recognition Research Network (FG-NET) ageing database [23]. The children were grouped as less than 12 years of age, adolescents aged 13 to 18 years, adults 19 to 59 years of age, and senior adults 60 years and above. In the AFDUC cohort, disease presentation was classified based on the symptoms observed by surveillance officers. As all cases were hospitalized, no cases were assigned as mild disease. Moderate disease was defined as having only febrile illness, and severe disease if at least one neurological syndrome, such as encephalitis, AFP, meningitis or seizures, were observed. Risk factors associated with WNV infection were calculated based on patient demographic and clinical features using a univariate odds ratio analysis, with associated 95.00% confidence intervals (CI), performed in EpiInfo (version 7.2.0.1) using Fisher’s exact test. 

## 3. Results

### 3.1. Neurological Cohort

A total of 219 archived CSF specimens submitted to the NHLS Tshwane Virology Laboratory between the months of January and June 2017 were tested for the presence of WNV. The patient’s demographic data is described in Table 1. Most of the CSF specimens tested were from children (116/219, 52.97%, 95.00% CI (46.13–59.73)), while the age group with the lowest submission was from adolescents (8/219, 3.65%, 95.00% CI (1.59–7.07)). Male (105/219, 47.95, 95.00% CI (41.17–54.78)) and female (106/219, 48.40%, 95.00% CI (41.62–55.23)) submissions were almost equal (8/219, 3.65%, 95.00% CI (1.59–7.07), patients had no gender information). Of the 219 CSF specimens, 2 (0.9%, 95.00% CI (0.11–3.26)) specimens (identifiers ZRUNH273/17 and ZRUNH497/17) tested positive on real-time RT-PCR for WNV. ZRUNH273/17 referred to a male child (1.5 years of age) who presented with meningitis in early March 2017. Eighteen days later in the same hospital, a six-month-old female child presented with similar symptoms and tested positive on the WNV real-time RT-PCR (identifier ZRUNH497/17). Sequencing and phylogenetic analysis of the 226 nucleotide (nt) region of the NS5 gene confirmed WNV lineage 2 infection in both cases (Figure 1). GenBank accession numbers for the WNV NS5 sequences are OL790166 and OL790167. The two sequences clustered with previously identified lineage 2 strains detected in mosquitoes and animals in SA. The pairwise distance analysis demonstrated very few nt differences (98.85–99.60%) in the NS5 gene region to highly neuroinvasive South African WNV strains isolated from humans and animals. West Nile virus IgM testing in combination with WNV VNT for confirmation detected an additional 6/219 (2.74%, 95.00% CI (1.01–5.87)) patients with acute infection. The two CSF specimens that tested WNV positive on real-time RT-PCR did not test positive for WNV IgM antibodies. Therefore, a total of 8/219 (3.65%, 95.00% CI (1.59–7.07)) patients with neurological symptoms in Gauteng hospitals in 2017 tested positive for WNV according to molecular and serodiagnosis on CSF (refer to Appendix A for details on PCR, IgM, and VNT test results). Percentage positivity peaked in January and March (Figure 2). The age range of WNV positive patients was six months to 61 years (median 10 years). The demographic risk factors and clinical symptoms associated with WNV infection are shown in Table 2. The majority of the WNV infections were identified in children (4/8, 50.00%, 95.00% CI (15.70–84.30)) and (6/8, 75.00%, 95.00% CI (34.91–96.81)) female patients. Presentations of encephalitis (2/8, 25.00%, 95.00% CI (3.19–65.09)), and febrile convulsions (1/8, 12.50%, 95.00 CI (0.32–52.65)) were common in the infected patients. The development of encephalitis was significantly more likely in WNV positive patients (OR = 34.88, *p* ≤ 0.05) when compared to WNV negative patients. There was no clear correlation between the age groups and the different neurological syndromes.

### 3.2. AFDUC Cohort

Routine molecular testing was performed on 979 AFDUC patients at both sentinel study sites for three years, yet no WNV positive cases were identified (refer to Appendix A for the demographic data). Acute WNV infections were investigated through IgM antibody testing on a subset of patients enrolled from January to June 2019 to 2021 (*n* = 441). The demographic data of the subset of patients selected for serological testing is shown in Table 3. Most cases were from 2019 (199/441, 45.12%, 95.00% CI (40.54–49.79)), children (312/441, 70.75%, 95.00% CI (66.34–74.80)), males (228/441, 51.70%, 95.00% CI (47.04–56.33)), and the Gauteng (268/441, 60.77%, 95.00% CI (56.14–65.22)) site. The details of the specimens tested and the associated results can be found in Table 4. A single specimen was tested for each patient, divided between 341/441 (77.32%, 95.00% CI (73.19–80.99)) blood specimens and 100/441 (22.68%, 95.00% CI (19.01–26.81)) CSF specimens. The majority of the CSF specimens tested (80/100, 80.00%, 95.00% CI (70.82–87.33)) arose from patients enrolled at the Gauteng site. From the 441 specimens investigated for WNV IgM antibodies, 61 (13.83%, 95.00% CI (10.92–17.37)) blood specimens and 2 (0.45%, 95.00% CI (0.12–1.64)) CSF specimens tested WNV IgM positive, totaling 63/441 (14.29%, 95.00% CI [11.33–17.86]) patients, however, only the 40 (9.07%, 95.00% CI (6.73–12.12)) patients whose infections could be confirmed by WNV VNTs were used for further analysis (refer to Appendix A for details on PCR, IgM and VNT test results). No neutralizing antibodies could be confirmed in the 2 CSF specimens. A total of 5.93% (16/270, 95.00% CI (3.42–9.45)) of patients presenting at the Gauteng sentinel site tested positive for WNV, while this increased to 14.04% (24/171, 95.00% CI (9.20–20.16)) in the Mpumalanga patients. The percentage of patients testing positive varied year-by-year, with 8.56% (16/187, 95.00% CI (4.97–13.52)) in 2019, 8.33% (12/144, 95.00% CI (4.38–14.10)) in 2020 and 10.91% (12/110, 95.00% CI (5.77–18.28)) in 2021. The peak antibody positivity was observed in February 2019 (3/22, 13.63%, 95.00% CI (2.91–34.91)) and March 2020 (4/26, 15.38%, 95.00% CI (4.36–34.87)) (Figure 2). Of the patients who tested WNV IgM positive but whose serum or plasma failed to neutralize WNV, no other flavivirus infections were identified through additional neutralization tests. 

The age range of WNV positive patients was six months to 90 years (median of 4 years). The demographic details of the patients who tested WNV positive are given in Table 5. The majority of WNV cases were identified in females (22/40, 55.00%, 95.00% CI (38.49–70.74)), children (20/40, 50.00%, 95.00% CI (33.80–66.20)), and patients from the rural Mpumalanga site (24/40, 60.00%, 95.00% CI (43.33–75.14)). The majority (29/40, 72.50%, 95.00% CI (56.11–85.40)) of infections were severe, based on at least one neurological symptom at the time of enrollment. Moderate disease was only observed in two age categories but was more commonly identified in children (7/11, 63.64%, 95.00% CI (30.79–89.07]) than in adults (4/11, 36.36%, 95.00% CI (10.93–69.21)), while severe disease was identified in every age category but was almost equally identified in children (13/29, 44.83%, 95.00% CI (26.45–64.31)) and adults (12/29, 41.38%, 95.00% CI (23.52–61.06)), followed by adolescents (3/29, 10.34%, 95.00% CI (2.19–27.35)) and senior adults (1/29, 3.45%, 95.00% CI (0.09–17.76)).

Table 6 describes the socio-demographic risk factors associated with WNV infection in patients presenting with AFDUC, with or without neurological disease, using odds ratio analysis. Age was the only risk factor identified in this study, with adolescent (OR = 5.34, 95.00% CI (1.28–22.22), *p* ≤ 0.05) and senior adult (OR = 1.69, 95.00% CI (0.20–14.38), *p* ≤ 0.05) patients significantly more at risk. Surprisingly, immunosuppression was not identified as a risk factor (OR = 0.91, 95.00% CI (0.39–2.13), *p* ≥ 0.05). Patients residing in the Mpumalanga province were twice as likely to test positive (OR = 2.54, 95.00% CI [1.31–4.92]) for WNV, but this was not statistically significant (*p* ≥ 0.05). Univariate analysis was performed on clinical signs to determine the association between WNV infection and severe neurological disease (Table 7). Headaches (OR = 2.18, 95.00% CI (1.31–4.20)), and neurological signs (OR = 1.88, 95.00% CI (0.91–3.87)) were significantly associated with WNV infection (*p* ≤ 0.05) in AFDUC patients. The most commonly observed neurological syndromes in WNV positive patients were meningitis (16/29, 55.17%, 95.00% CI (35.69–73.55)), AFP (6/29, 20.69%, 95.00% CI (7.99–39.72)), seizures (3/29, 10.34%, 95.00% CI (2.19–27.35)) and encephalitis (1/29, 3.45%, 95.00% CI (0.09–17.76)), but the development of these syndromes was not statistically associated with infection (*p* ≥ 0.05). Of the 263 AFDUC patients experiencing neurological syndromes, WNV was identified in 29 (11.03%, 95.00% CI (7.51–15.45)). A one-month follow-up was performed on 345/441 (78.23%, 95.00% CI (74.14–81.83)) patients, and from those who were WNV-positive (*n* = 32), a recovery rate of 84.38% (27/32, 95.00% CI (67.21–94.72)) was identified. The remaining 15.62% (5/32, 95.00% CI (5.28–32.79)) were still symptomatic. A single death was recorded (CFR = 2.50%).

## 4. Discussion

Despite originating in Africa, limited data exist on the burden of WNV disease in humans on the African continent. Given the global burden of WNV, there is an increasing need for improved information on the phylogeny, epidemiology, and circulation on the African continent [24]. Currently, no studies determining demographic and socioeconomic risk factors for WNV infection in SA exist, nor does a study examining a large group of hospitalized patients experiencing acute fever of unknown cause with or without neurological signs. In this study, investigation of CSF specimens from hospitalized patients experiencing acute neurological disease in Gauteng in 2017 identified WNV in 8/219 (3.65%, 95.00% CI (1.59–7.07)) patients, which suggests a higher incidence than a smaller study focusing on a similar population in South Africa [15]. Phylogenetic analysis confirmed endemic lineage 2 strains circulating between mosquitoes, animals, and humans in SA. The interannual WNV detection rate in patients experiencing acute febrile disease with or without neurological symptoms at two sentinel sites in the Gauteng and Mpumalanga provinces ranged from 8.33–10.91% from 2019 to 2021. This detection rate is in line with that noted in horses in SA [18]. The variability in detection rates could be due to annual changes in rainfall or changes in behavior during the COVID-19 pandemic. This high detection rate supports the notion that WNV is underestimated in SA [9]. The presence of clinical disease that matches the case definition for arboviral infections, along with IgM antibodies confirmed with neutralization tests, suggests these are likely acute WNV infections, despite the known persistence of these antibodies. 

Since most WNV disease occurs in adults [25], the true burden of WNV disease in SA may be underestimated as the median age in both cohorts was 10 and 4 years of age. Few other studies included young children as part of the study group. Moderate and severe disease was most noted in children compared to any other age category. The high detection of IgM antibodies in children and the detection of WNV lineage 2 in two children with neurological signs suggest that WNV should be considered in neurological cases in children. Despite over half the infections being identified in children, adolescent and senior adult patients had a higher risk of having WNV infection, although these statistics are likely affected by the high number of children enrolled into the ANDEMIA study. This bias could also explain why in contrast to other studies, disease was not associated with a compromised immune system due to HIV, as children are less likely to be HIV positive. The higher detection rate at the rural Mpumalanga site may be due to a higher abundance of mosquito breeding sites, amplification hosts, and increased contact between humans and mosquitoes through a more outdoor lifestyle and increased farming activity compared to the patients at the urban sentinel site. There is a geographical correlation between areas where WNV has been identified in animals and mosquitoes, suggesting areas of endemic transmission between mosquitoes, animals, and humans within the country [17,18]. 

West Nile virus accounted for 11.03% (29/263, 95.00% CI (7.51–15.45)) of AFDUC cases that presented with at least one neurological symptom in hospitals in Gauteng and Mpumalanga province from 2019 to 2021, and 72.50% (29/40, 95.00% CI (56.11–85.40)) of all WNV cases had a neurological component, suggesting WNV may contribute significantly to unsolved cases of neurological disease in hospitals in SA. The prevalence of AFP in WNV cases is variable but usually ranges between 5–15% [26], which is lower than the results presented in this study (6/29, 20.69%, 95.00% CI (7.99–39.72)). However, AFP may be more common in younger patients [27]. This may explain the high percentage of AFP in the study, as the majority of the patients enrolled were children. A high recovery rate was observed, which is in concordance with previous studies [2]. A single death was recorded in a 38-year-old HIV positive male who, at the time of death, was experiencing sepsis and AFP. 

Although no PCR positive cases and phylogenetic data could not be obtained for the WNV positive patients in the AFDUC cohort (2019 to 2021), lineage 2 is well known to be the most common lineage circulating in SA and was identified in 2 PCR positive cases identified in the neurological CSF cohort from 2017. Syndromic surveillance of horses and other animals [21] from 2017–2020 also detected WNV lineage 2 in all cases in the same regions except a single lineage 1 case identified in a lion in the Kruger National Park in 2017. Molecular testing may be a suitable diagnostic tool during early infection, but after seroconversion has taken place, serological testing is more likely to correctly identify cases. However, VNTs are required for confirmation for accurate epidemiological studies. The number of confirmed WNV cases reduced from 14.29% (63/441, 95.00% CI (11.33–17.86)) to 9.07% (40/441, 95.00% CI (6.73–12.12)) using VNTs. We were unable to identify if another flavivirus was responsible for these cases despite testing for co-circulating flaviviruses using VNTs. The detection of WNV IgM antibodies in serum or plasma shows a CSF specimen is not crucial for diagnosis. 

This study has a few limitations, the first being the high enrolment numbers of children in the AFDUC cohort. The low enrolment numbers in the other age groups make the accurate determination of the relative proportion difficult. Secondly, IgM ELISA was used for screening of serum and CSF samples, however not all neurological cases had CSF specimens available and were diagnosed on serum alone. As CSF was preferentially tested over serum for WNV IgM, fewer neuroinvasive cases may have been missed. Both serum and CSF should be tested if available for a single patient. Since we did not have paired serum samples that would show an increase in antibody levels, we only considered IgM-positive cases that could be confirmed via virus neutralization tests as positive. As we only tested patient sera or CSF that failed to neutralization WNV for cross-reactivity to other flaviviruses after the WNV VNTs were performed, this leaves the possibility that a some WNV VNT results were false positives, as cross-reactivity needs to be performed simultaneously, and the decision reached is based on which titer is the highest. 

There was limited access to patients placed in the intensive care unit, and patients who had unfortunately demised were unable to give consent. Therefore, the enrolments of patients experiencing severe AFDUC symptoms were likely affected.

To conclude, the findings identified WNV as the etiological agent in 8–11% of hospitalized patients experiencing acute febrile and/or neurological disease in SA. This study lays the groundwork for defining the true burden of WNV amongst undiagnosed cases of febrile and neurological disease in hospitalized patients in South Africa and suggests a significant disease burden not previously known for Africa. Our study confirms that WNV should form part of the differential diagnoses of AFDUC cases with neurological signs in hospitalized patients in SA, particularly in children. Increased clinical awareness and prospective surveillance across countries in which animal cases were previously detected as well as in other parts of Africa is needed to define the true burden. 

## Figures and Tables

**Figure 1 viruses-15-02207-f001:**
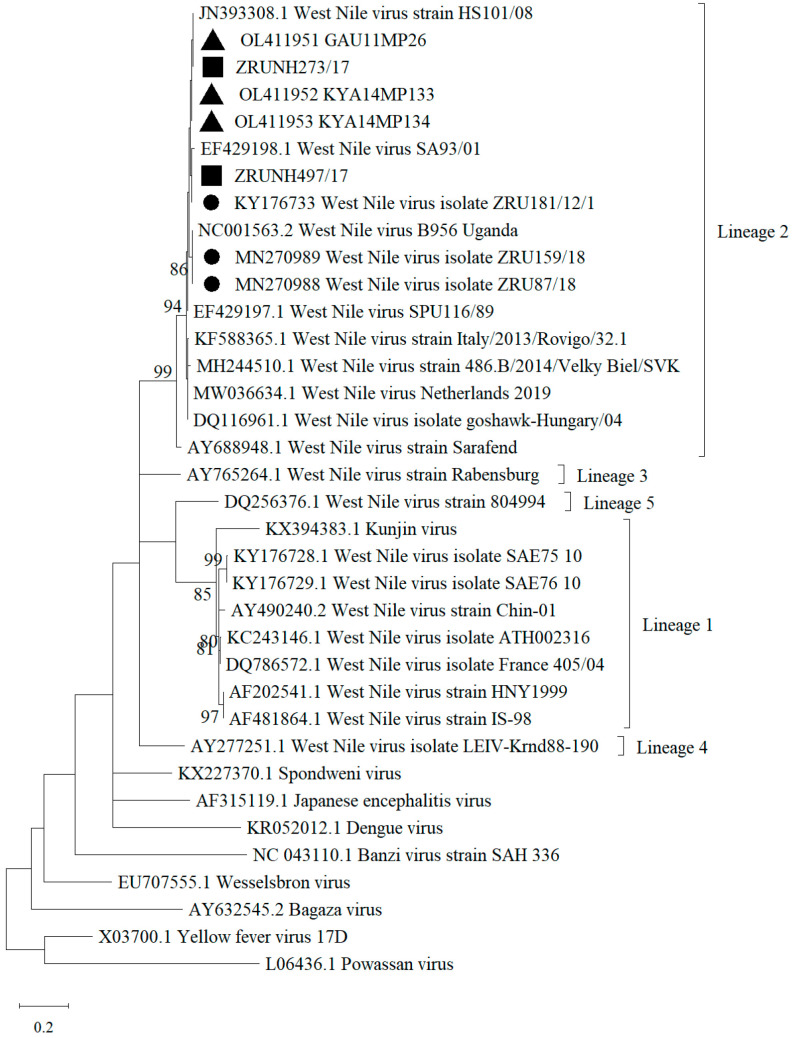
Maximum Likelihood phylogram of the partial NS5 gene (nt = 226) used for confirmation of WNV human infection indicating bootstrap (bs) support values on the branches (36 taxa). Viral sequences obtained in this study from hospitalized patients in SA are indicated by filled squares. Filled black triangles indicate sequences identified in mosquito vectors [15]. Filled circles represent the viral sequences identified in animals in South Africa [18]. Values are shown if bs support was ≥70. The evolutionary history was inferred using the Kimura 2-parameter model plus gamma distribution. The tree is drawn to scale and branch lengths represent the number of nucleotide substitutions per site. Reference sequences were downloaded from GenBank. GenBank accession numbers for the WNV NS5 sequences are ZRUNH273/17 = OL790166, ZRUNH497/17 = OL790167.

**Figure 2 viruses-15-02207-f002:**
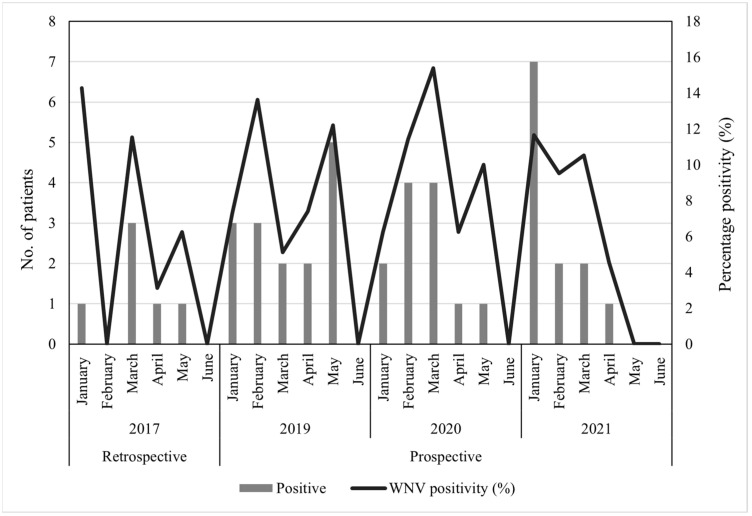
The seasonality of WNV in hospitalized patients in South Africa, January–June 2017, 2019 to 2021.

**Table 1 viruses-15-02207-t001:** Demographic data of the 219 cerebrospinal fluid specimens retrospectively tested for West Nile virus in South Africa, 2017.

Category	Frequency	Percentage (%) (95.00% CI)
Age		
Children (0–12 years)	116/219	52.97 (46.13–59.73)
Adolescents (13–18 years)	8/219	3.65 (1.59–7.07)
Adults (19–59 years)	79/219	36.07 (29.71–42.82)
Senior adults (≥60 years)	9/219	4.11 (1.90–7.66)
Unknown	7/219	3.20 (1.29–6.47)
Sex		
Female	106/219	48.40 (41.62–55.23)
Male	105/219	47.95 (41.17–54.78)
Unknown	8/219	3.65 (1.59–7.07)

**Table 2 viruses-15-02207-t002:** Risk factors associated with West Nile virus infection and clinical symptoms/syndromes in hospitalized patients with neurological disease in South Africa, January to June 2017.

	West-Nile-Virus-Positive (*n* = 8) (%)	West-Nile-Virus-Negative (*n* = 211) (%)	Odds Ratio (95.00% Confidence Interval)	*p*-Value
Age				
Children (0–12 years)	4/8 (50.00)	112/211 (53.08)	0.88 [0.22–3.63]	1.00
Adolescents (13–18 years)	0/8 (0.00)	5/211 (2.37)	0 (undefined)	1.00
Adults (19–59 years)	3/8 (37.50)	76/211 (36.02)	1.06 (0.25–4.58)	1.00
Senior adults (≥60 years)	1/8 (12.50)	9/211 (4.27)	3.20 (0.36–28.91)	0.32
Unknown	0/8 (0.00)	9/211 (4.27)	0 (undefined)	1.00
Sex				
Female	6/8 (75.00)	100/211 (47.39)	3.33 (0.66–16.87)	0.16
Male	2/8 (25.00)	111/211 (52.61)	0.30 (0.06–1.52)	0.16
Clinical Symptoms/Syndromes				
Encephalitis	2/8 (25.00)	2/211 (0.95)	34.88 (4.17–290.60)	0.01
Febrile convulsions	1/8 (12.50)	7/211 (3.32)	4.16 (0.44–38.59)	0.26
Meningitis	4/8 (50.00)	52/211 (24.64)	3.06 (0.73–12.66)	0.21
Encephalopathy	1/8 (12.50)	2/211 (0.95)	14.93 (1.21–184.78)	0.11

**Table 3 viruses-15-02207-t003:** Demographic data of the subset of AFDUC patients enrolled for West Nile virus serological testing, January to June 2019 to 2021.

	Gauteng (*n* = 268)		Mpumalanga (*n* = 173)		Total (*n* = 441)	
	Frequency	Percentage (%) (95.00% CI)	Frequency	Percentage (%) (95.00% CI)	Frequency	Percentage (%) (95.00% CI)
Age						
Children (0–12 years)	214/268	79.85 (74.54–84.49)	98/173	56.65 (48.91–64.15)	312/441	70.75 (66.34–74.80)
Adolescents (13–18 years)	2/268	0.75 (0.09–2.67)	6/173	3.47 (1.28–7.40)	8/441	1.81 (0.92–3.54)
Adults (19–59 years)	51/268	19.03 (14.51–24.25)	64/173	36.99 (29.79–44.65)	115/441	26.08 (22.20–30.37)
Senior adults (≥60 years)	1/268	0.37 (0.01–2.06)	5/173	2.89 (0.94–6.62)	6/441	1.36 (0.62–2.94)
Sex						
Male	138/268	51.49 (45.33–57.62)	90/173	52.02 (44.31–59.66)	228/441	51.70 (47.04–56.33)
Female	130/268	48.51 (42.38–54.67)	83/173	47.98 (40.34–55.69)	213/441	48.30 (43.67–52.96)
Total patients screened						
2019	121/268	45.15 (39.09–51.32)	78/173	45.09 (37.52–52.82)	199/441	45.12 (40.54–49.79)
2020	73/268	27.24 (22.00–32.99)	53/173	30.64 (23.86–38.08)	126/441	28.57 (24.55–32.96)
2021	74/268	27.61 (22.35–33.38)	42/173	24.28 (18.09–31.37)	116/441	26.30 (22.41–30.61)

**Table 4 viruses-15-02207-t004:** Details of the West Nile virus testing, including number of AFDUC patients and specimens tested, January to June 2019 to 2021.

	Gauteng			Mpumalanga			Total (*n* = 441)
Specimens Tested per Year	2019 (*n* = 112)	2020 (*n* = 88)	2021 (*n* = 70)	2019 (*n* = 75)	2020 (*n* = 56)	2021 (*n* = 40)	
Type of specimen tested (tested/total specimens tested)							
Blood (%) (95.00% CI)	102/112 (91.07) (84.19–95.64)	45/88 (51.14) (40.25–61.95)	43/70 (61.43) (49.03–72.83)	64/75 (85.33) (75.27–92.44)	48/56 (85.71) (73.78–93.62)	39/40 (97.50) (86.84–99.94)	341/441 (77.32) (73.19–80.99)
CSF (%) (95.00% CI)	10/112 (8.93) (4.36–15.81)	43/88 (48.86) (38.05–59.75)	27/70 (38.57) (27.17–50.97)	11/75 (14.67) (7.56–24.73)	8/56 (14.29) (6.38–26.22)	1/40 (2.50) (0.06–13.16)	100/441 (22.68) (19.01–26.81)
Test results							
Blood IgM positive (%) (95.00% CI)	8/112 (7.14) (3.13–13.59)	8/88 (9.09) (4.01–17.13)	8/70 (11.43) (5.07–21.28)	13/75 (17.33) (9.57–27.81)	10/56 (17.86) (8.91–30.40)	14/40 (35.00) (20.63–51.68)	61/441 (13.83) (10.92–17.37)
CSF IgM positive (%) (95.00% CI)	0/112 (0.00) (0.00–3.24)	0/88 (0.00) (0.00–4.11)	0/70 (0.00) (0.00–5.13)	0/75 (0.00) (0.00–4.80)	1/56 (1.79) (0.05–9.55)	1/40 (2.50) (0.06–13.16)	2/441 (0.45) (0.12–1.64)
Combined positive (%) (95.00% CI)	8/112 (7.14) (3.13–13.59)	8/88 (9.09) (4.01–17.13)	8/70 (11.43) (5.07–21.28)	13/75 (17.33) (9.57–27.81)	11/56 (19.64) (10.23–32.43)	15/40 (37.50) (22.73–54.20)	63/441 (14.29) (11.33–17.86)
Confirmed through VNT	8/112 (7.14) (3.13–13.59)	6/88 (6.82) (2.54–14.25)	2/70 (2.86) (0.35–9.94)	8/75 (10.67) (4.72–19.94)	6/56 (10.71) (4.03–21.88)	10/40 (25.00) (12.69–41.20)	40/441 (9.07) (6.73–12.12)

**Table 5 viruses-15-02207-t005:** Details of the West Nile virus positive AFDUC patients in South Africa. January to June 2019 to 2021.

	Gauteng (*n* = 16)	Mpumalanga (*n* = 24)	Total (*n* = 40)
Sex			
Male (%) (95.00% CI)	5/16 (31.25) (11.02–58.66)	13/24 (54.17) (32.82–74.45)	18/40 (45.00) (29.26–61.51)
Female (%) (95.00% CI)	11/16 (68.75) (41.34–88.98)	11/24 (45.83) (25.55–67.18)	22/40 (55.00) (38.49–70.74)
Age			
Children (0–12 years) (%) (95.00% CI)	15/16 (93.75) (69.77–99.84)	5/24 (20.83) (7.13–42.15)	20/40 (50.00) (33.80–66.20)
Adolescents (13–18 years) (%) (95.00% CI)	0/16 (0.00) (0.00–20.59)	3/24 (12.50) (2.66–32.36)	3/40 (7.50) (1.57–20.39)
Adults (19–59 years) (%) (95.00% CI)	1/16 (6.25) (0.16–30.23)	15/24 (62.50) (40.59–81.20)	16/40 (40.00) (24.86–56.67)
Senior adults (≥60 years) (%) (95.00% CI)	0/16 (0.00) (0.00–20.59)	1/24 (4.17) (0.11–21.12)	1/40 (2.50) (0.06–13.16)
Disease Presentation			
Moderate (%) (95.00% CI)	7/16 (43.75) (19.75–70.12)	4/24 (16.67) (4.74–37.38)	11/40 (27.50%) (14.60–43.89)
Severe (%) (95.00% CI)	9/16 (56.25) (29.88–80.25)	20/24 (83.33) (62.62–95.26)	29/40 (72.50) (56.11–85.40)

**Table 6 viruses-15-02207-t006:** Demographic and socioeconomic factors associated with West Nile virus infection in AFDUC patients in South Africa, January to June 2019 to 2021.

	West-Nile-Virus-Positive (*n* = 40) (%)	West-Nile-Virus-Negative (*n* = 401) (%)	Odds Ratio (95.00% CI)	*p*-Value
Sex				
Female	22/40 (55.00)	191/401 (47.63)	1.34 (0.70–2.58)	0.41
Male	18/40 (45.00)	210/401 (52.37)	0.74 (0.39–1.43)	0.41
Age				
Children (0–12 years)	20/40 (50.00)	289/401 (72.07)	0.39 (0.21–0.75)	0.01
Adolescents (13–18 years)	3/40 (7.50)	6/401 (1.50)	5.34 (1.28–22.22)	0.05
Adults (19–59 years)	16/40 (40.00)	100/401 (24.94)	2.00 (1.03–3.93)	0.33
Senior adults (≥60 years)	1/40 (2.50)	6/401 (1.50)	1.69 (0.20–14.38)	0.02
Sentinel site				
Gauteng	16/40 (40.00)	252/401 (62.84)	0.39 (0.21–0.77)	0.00
Mpumalanga	24/40 (60.00)	149/401 (37.16)	2.54 (1.31–4.92)	0.71
HIV status				
Negative	26/40 (65.00)	288/401 (71.82)	0.73 (0.37–1.45)	0.23
Positive	7/40 (17.50)	76/401 (18.95)	0.91 (0.39–2.13)	0.50
Unknown	7/40 (17.50)	37/401 (9.23)	2.01 (0.86–5.05)	0.10

**Table 7 viruses-15-02207-t007:** Clinical symptoms/syndromes, recovery rate, and case fatality rate associated with WNV infection in AFDUC patients in South Africa, 2019 to 2021.

Disease Presentation	West-Nile-Virus-Positive (*n* = 40) (%)	West-Nile-Virus-Negative (*n* = 401) (%)	Odds Ratio (95.00% CI)	*p*-Value
Swollen lymph nodes	2/40 (5.00)	3/401 (0.75)	6.98 (1.13–43.01)	0.07
Sepsis	3/40 (7.50)	10/401 (2.49)	3.17 (0.84–12.03)	0.10
Arthralgia	4/40 (10.00)	27/401 (6.73)	1.54 (0.51–4.64)	0.51
Headache	20/40 (50.00)	126/401 (31.42)	2.18 (1.31–4.20)	0.02
Neurological	29/40 (72.50)	234/401 (58.35)	1.88 (0.91–3.87)	0.05
Neurological	West-Nile-virus-positive (*n* = 29) (%)	West-Nile-virus-negative (*n* = 234) (%)	Odds ratio (95.00% CI)	*p*-value
Encephalitis	1/29 (3.45)	2/234 (0.85)	4.14 (0.36–47.17)	0.30
Seizures	3/29 (10.34)	35/234 (14.96)	0.66 (0.19–2.29)	0.78
Acute flaccid paralysis	6/29 (20.69)	51/234 (21.79)	0.94 (0.36–2.42)	1.00
Meningitis	16/29 (55.17)	102/234 (43.59)	1.59 (0.73–3.46)	0.24
One month follow-up	West-Nile-virus-positive (*n* = 32) (%)	West-Nile-virus-negative (*n* = 313) (%)	Odds ratio (95.00% CI)	*p*-value
Alive	31/32 (96.88)	308/313 (98.40	0.50 (0.06–4.45)	0.44
Dead	1/32 (3.13)	5/313 (1.60)	1.99 (0.22–17.55)	0.44
Recovered	27/32 (84.38)	266/313 (84.98)	0.95 (0.34–2.60)	1.00

## Data Availability

Sequencing data presented in this study are openly available on GenBank using the following accession numbers: OL790166, OL790167. Patient data presented in this study are available on request from the corresponding author. The data are not publicly available due to privacy reasons.

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
