# Peer review of "West Nile Virus, an Underdiagnosed Cause of Acute Fever of Unknown Origin and Neurological Disease among Hospitalized Patients in South Africa"

_viruses, 2023, doi:10.3390/v15112207_

Round 1
Reviewer 1 Report
Comments and Suggestions for Authors
In this manuscript entitled West Nile Virus, a missed cause of acute fever of unknown cause and neurological disease among hospitalized patients in South Africa, the authors show that West-Nile virus is an actual cause of neurological and febrile diseases in hospitalized patients in South Africa. This work highlights concerns about WNV circulation and public health importance for diagnosis of this pathogen especially in the context of unsolved acute fever and neurological disease of unknown origin.
The main points concerned for revision are more precision on results and demographic data.
Have the 2 positive specimens for RT-qPCR (from neurological cohort) also been tested for serological and VNT? If not, why? Should be interesting to have correlation between PCR results (and quantification), IgM level and VNT.
Results shown in tables are clear and understandable but, authors should include data to show sex ratio under each age group, but also severity and symptoms (table 2, 5 and 6). E.g. Are encephalopathies more current in children?
Table 4. Concerning the second cohort, why weren’t tests done when both blood and CSF samples were available? As you also mentioned that CSF is not crucial for diagnosis (lines 342-343), it could be interesting to have results for these patients in a separate table (e.g. patient X found positive for IgM in blood AND CSF…).
Concerning VNT, did you realize and test serum dilution? Why do not include VNT antibodies titer?
Minor
Title: could be improved, it contains two times the term cause; e.g. “unknow origin…”
Methods: a brief summary of WNV VNT method could be added
Line 236 : mistake in antibodies spelling
Reviewer 2 Report
Comments and Suggestions for Authors
No modifications necessary
Author Response
Thank you for your critical review of the manuscript.
Reviewer 3 Report
Comments and Suggestions for Authors
Dear Authors,
I have thoroughly read the manuscript named „West Nile Virus, a missed cause of acute fever of unknown 2 origin and neurological disease among hospitalized patients in South Africa”. I find the thematics important and results interesting.
Please find the point by point comments in the text below:
TITLE:
I suggest changing the title to WNV, an underdiagnosed cause of acute fever and neurological disease
Line 18. Please add confirmation of serological resuts and used molecular testing…
Line 22. Please write also the numbers.
Line 22.25. and throughout the text. Please include confidence intervals
Line 25. Please shortly state which neurological signs
Line 44. Please change the sentence in more
Line 166. Typo TCID
Line 161-173. This paragraph can be avoided.
Line 174. Comment: Usually cross-reactivity needs to be done simultaneously since the decission on cross reactivity depends on which titre was higher. This approach leaves possibility that a part of positive results declared for WNV is still cross reaction.
Table S1. Please add GenBank to accesion No title
Comments on the Quality of English LanguageMinor English language editing is necessary.
